# A study on tuberculosis disease disclosure patterns and its associated factors: Findings from a prospective observational study in Chennai

**Karikalan Nagarajan**[1], **Malaisamy Muniyandi**[2]*, **Senthil Sellappan**[3], **Srimathi Karunanidhi**[1], **Keerthana Senthilkumar**[1], **Bharathidasan Palani**[4], **Lavanya Jeyabal**[5], **Rajendran Krishnan**[6]

1 Department of Social and Behavioral Research, ICMR-National Institute for Research in Tuberculosis, Chennai, India, 2 Department of Health Economics, ICMR–National Institute for Research in Tuberculosis, Chennai, India, 3 ICMR-Regional Medical Research Centre, Port Blair, Andaman and the Nicobar Islands, India, 4 Department of Statistics, ICMR–National Institute for Research in Tuberculosis, Chennai, India, 5 District TB Office, National TB Elimination Programme, Chennai, India, 6 Department of Statistics (Epidemiology Unit), ICMR–National Institute for Research in Tuberculosis, Chennai, India

* muniyandi.m@icmr.gov.in

**Data Availability Statement:** Data use for this study is available from the Institutional Ethics Committee of ICMR-NIRT for researchers who

## Abstract

### Background

Disclosure of tuberculosis (TB) status by patients is a critical step in their treatment cascade of care. There is a lack of systematic assessment of TB disclosure patterns and its positive outcomes which happens dynamically over the disease period of individual patients with their family and wider social network relations.

### Methods

This prospective observational study was conducted in Chennai Corporation treatment units during 2019–2021. TB patients were recruited and followed-up from treatment initiation to completion. Information on disease disclosures made to different social members at different time points, and outcomes were collected and compared. Bivariate and multi variate analysis were used to identify the patients and contact characteristics predictive of TB disclosure status.

### Results

A total of 466 TB patients were followed-up, who listed a total of 4039 family, extra familial and social network contacts of them. Maximum disclosures were made with family members (93%) and half of the relatives, occupational contacts and friendship contacts (44–58%) were disclosed within 15 days of treatment initiation. Incremental disclosures made during the 150–180 days of treatment were highest among neighbourhood contacts (12%), and was significantly different between treatment initiation and completion period. Middle aged TB patients (31 years and 46–55 years) were found less likely to disclose (AOR 0.56 and 0.46 respectively; p<0.05) and illiterates were found more likely to disclose their TB status

meet the criteria for access to confidential data. Corresponding Author could be communicated in this regard. Public data sharing is not possible for this study publicly because of patient confidentiality who participated in this study. Correspondence could be made to nirtiec@gmail.com.

**Funding:** The Indian Council of Social Science Research (ICSSR), Ministry of Human Resource Development, Government of India (No. IMPRESS/ 2588/462/2018-19/ICSSR) through the Impactful Policy Research in Social Science (IMPRESS) Scheme.

**Competing interests:** The authors declare that no competing interest exists.

(AOR 3.91; p<0.05). Post the disclosure, family contacts have mostly provided resource support (44.90%) and two third of all disclosed contacts have provided emotional support for TB patients (>71%).

## Conclusion

Findings explain that family level disclosures were predominant and disclosures made to extra familial network contacts significantly increased during the latter part of treatment. Emotional support was predominantly received by TB patients from all their contacts post disclosure. Findings could inform in developing interventions to facilitate disclosure of disease status in a beneficial way for TB patients.

## Background

An estimated 1.8 million people are affected by tuberculosis (TB) every year in India [1]. TB remains a major public health problem with significant burden for the health system [2]. TB is strongly influenced by the social factors like poverty, stigma, social environment and cultural factors [3]. Treating TB and its related psychosocial and economic consequences requires significant familial, social and livelihood support in addition to the standard and free medical care provided by the National TB Elimination Program (NTEP) in India [4, 5]. TB patients remain less supported, negatively perceived and are stigmatized at family, community and institutional settings [6, 7]. Self-disclosure of patients about their disease status could lead to patient neglect, withdrawal of familial and social support due to the stigma, negative perceptions and myths surrounding TB [8–10]. Alternatively, self-disclosure of patients about their disease status could enable the needed support for them by their family and social network members [8]. Disclosure of TB status could facilitate appropriate infection control measures and access to diagnostic, curative and preventive TB services by the patient's their close contacts.

Few studies in the past have attempted to assess the disclosure status of TB patients and their post disclosure experiences in India. A cross sectional study conducted in South India found that, among TB patient stigmatizing and negative experiences were faced by 10–25% of their family members, 54% from their community members, almost 35% of the TB patients lost their jobs or changed their jobs following disclosure [11]. A qualitative finding from South India identified stronger association of psycho social problems and disease disclosure among TB patients [12]. In terms of socio-economic status and gender, findings from a study conducted in North India found that 60% of the patients who belong to better socio-economic strata have hided their disease from close friends or neighbors and also suffered severe stigma. The study also indicated that women were more prone to disclose the disease status with their friends and colleagues but they were also facing higher stigma [13]. These Indian studies from different socio cultural settings highlighted the different patterns in the disclosure status of TB patients and the generally negative experiences which they received in the form of stigma. Similar studies in other countries have identified that non-disclosure of disease being practiced by TB patients to avoid stigma and discrimination. Its notable that disclosure and stigma in turn could result in low level of self-esteem, isolation depression, poor quality of life, barriers to understanding disease progress, treatment and prevention interventions [14–16].

Few studies have assessed the geographic and cultural variation in TB disclosure. TB patients in urban settings were more likely to feel ashamed and embarrassed and almost half of

them perceived community rejection as an outcome of disclosure [17, 18]. TB disclosure and its impact on sexual relationships were explored in a study from Bangladesh which found that disclosure to the spouse resulted in abstention from sexual relationship and non-disclosure lead to forceful relationship [19]. The effects of non-disclosure on TB transmission was underscored by a study conducted in Ethiopia which found that complete non-disclosure of TB status increased the likelihood of drug resistance status [20, 21].

Inspite of availability of multiple studies about TB disclosure, what remains a gap in the research literature is the lack of a systematic assessment of the disclosure patterns in relation to the time and person with whom the disclosures were made. While published studies have measured disclosure using a cross-sectional approach, assessment of disclosure requires a longitudinal follow up, which could track the disclosure status dynamically over the active disease period of individual TB patients. Disclosure patterns also requires assessment in the backdrop of specific family and other social network relations of the individual patients. So far studies have not collected data on the specific family and social network members of patients and their characteristics. Also studies in general have assessed the negative consequences of disclosure but the positive outcome of disclosure had not been documented. A recent study conducted in Uganda had documented that most TB patients had received positive support in the form of motivations and financial support after disease disclosure, thus underscoring the importance of assessing TB disclosure from a positive perspective [22]. In this backdrop, we undertook a longitudinal study to assess the disclosure patterns of TB patients within their familial and social network relationships at different time points during their treatment period.

Research questions of the study: We specifically aimed to assess the following questions ii) Measure and assess the significant difference in the proportions of disclosures made by individual TB patients to their family and social network members at their time of diagnosis, treatment continuation and completion; ii) to assess the background characteristics of TB patients and their contacts which were predictive of their TB disclosure and non-disclosure status; and iii) to measure the different types of post-disclosure supports which were received by the TB patients from their contacts.

## Methodology

### Study setting

Chennai is a metropolitan city of South India with a population of 8 million. Each year on an average seven thousand TB cases are diagnosed and notified under the NTEP program of Chennai, which provides diagnostic and treatment services through its Designated Microscopy Centers and Treatment Units. Patients are diagnosed and treated as per the NTEP guidelines [23].

### Study design

This study utilized a prospective observational design in which TB patients were followed-up from the time point of their treatment initiation till treatment completion at the end of six months or till treatment extension period to collect quantitative data on their TB disclosure status.

### Inclusion criteria

Adult (>18 years) newly diagnosed drug-sensitive pulmonary and extra pulmonary TB patients who were recruited from the diagnostic and treatment facilities of the NTEP of Greater Chennai Corporation during 2019–2021.

## Exclusion criteria

Drug-resistant TB patients, TB patients with HIV co-infection and other life-threatening comorbidity were excluded.

## Sample size

Sample size was calculated for the study based on the previous reported disclosure and non-disclosure status of TB in the study area [11], and with statistical assumptions which would allow to test significant difference in patient characteristics based on the disclosure status. Assuming confidence interval of 95%, precision of 20% relative precision and accounting 15 to 20% loss-to-follow-up, the minimum required sample size was estimated as 480 for a single site in Chennai. The formula used for calculating the sample size was $n = Z^2 p (1-p)/d^2$, in which parameters Z, p and d indicates confidence intervals, prevalent level of disclosures and desired precision respectively.

## Sampling methods

Probability proportional to size (PPS) sampling method was used to recruit samples from the treatment units (TUs) of Chennai based on caseloads. The PPS method was used to sample the patients from the 36 TUs, which have recorded a varying number of cases. This method was used to ensure that the probability of selecting a TB treatment unit is consistent with the volume of patients who are registered. The TUs in Chennai have disproportionate caseloads, and the PPS method provided the flexibility for selecting both larger and smaller volume TUs with respectively higher and lower probability respectively. A sampling frame was created using the TB caseload in the TUs for the previous year and eligible patients were consecutively recruited using the TB treatment registers. Patients were selected from TUs until saturation of the sample was reached. In case samples from a TU couldn't be reached completely due to operational reasons, the next nearby TU was selected.

## Operational definitions used for the study

**Complete disclosure.**   The patients disclosing his/her TB disease status to anyone of his family or social network contacts which is complete and voluntary.

**Non-disclosure.**   The patients complete or partial non-disclosure of his/her TB disease status to anyone of his family or social network contacts.

**Social networks of TB patients.**   Individuals with whom the TB patients had socialized, worked, stayed together, repeatedly shared information's and materials, repeatedly sought support or guidance in the past one year.

**Resources support.**   Provision of any monetary and non-monetary support (including money, food, special nutrition, etc.) by the listed family or social network members to the TB patients which helped them during the treatment period.

**Instrumental support.**   Provision of practical support (could be: taking to hospital, helping in household works, supporting in childcare, providing medicines on time, supporting others in day-to-day needs) by the listed family or social network members to the TB patients which helped them during the treatment period.

**Psychological or emotional support.**   Provision of motivation or counselling by the listed family or social network members to the TB patients which helped them during the treatment period to cope up when they felt psychologically down, stressed or depressed.

## Data collection process

Data collection was done by trained social workers who used printed questioners. Data collectors prospectively collected the list of newly diagnosed drug-sensitive TB patients (pulmonary and extra pulmonary) in the treatment units of study area using the NTEP treatment registers. Eligible participants were approached for participation in this study and ethical consent were obtained. Further, the following study related information were collected using pre-coded, semi-structured questionnaire. This included demographic and background characteristics of patient including age, gender, education, occupation and marital status at baseline. Information on the disclosure status of TB patients was inquired at three points at treatment initiation (within 15 days after treatment initiation), end of intensive phase of treatment (60–90 days after treatment initiation) and end of treatment completion (150–180 days after treatment initiation). Information on family and social network contacts of patients were collected using the standard name generators methods used in ego-centric network surveys [4]. Participants were required to free list their social network contacts (by using their name or nickname) and were asked about their relationship status with the listed member. The network relations were broadly categorised as family contacts, extended family/relative contacts, friends, occupational and neighbourhood contacts. Contacts background information including age, gender, education and occupational status. Information on the type of benefits received (if any) by the patients from different social network contacts post their TB disclosure was elicited as per the study definition.

## Data validity and tool standardisation

To ensure consistency and accuracy of the disclosure and social network data collected from patients we ensured the following steps a) respondent free listing of social network contacts, b) test-retest methods to check consistency of responses c) cross-checking of responses between different time points of survey and d) measures to gain respondents confidence with interviewers [4].

## Statistical analysis

All data were entered in excel and was analysed using STATA version 15.1. Descriptive statistics with frequency and proportions were used to characterize the background characteristics of the participants. Proportions with 95% confidence intervals were calculated to measure TB disclosure status and post disclosure support. To determine the significant difference in proportions of disclosure status Z test with the p-value significance at $<0.05$ was used. To determine the association between participant characteristics, disclosure status and supports received, Chi-Square test with p-value significance at $<0.05$ was used. Univariate and multivariate logistic regression analysis was used to calculate unadjusted and adjusted odds ratio for variables which were predictive of disclosure status. Variables significantly associated with disclosure status in univariate model were only assessed for significance in multivariate models with p-value significance of $<0.05$. To understand the multicollinearity of the variables included in the multiple regression we calculated individual variance inflation factors (VIF) for each and mean VIA. A threshold of $<4$ was considered as acceptable level of collinearity [24].

**Dependent and independent variables for regression analysis.** The dependent variable was constructed as a categorical variable which was coded as 1 if TB disclosure was 'yes' and 0 if disclosure was 'no'. TB patient's age gender, occupation, marital status, education status and TB type were used as independent categorical variables. Contact characteristics including age, gender, occupation, educational status and relation type were used as independent categorical variables.

## Ethical considerations

This study was approved by the Institutional Ethic Committee of ICMR-NIRT (NIRT-IEC: 2019007) and was implemented between September 2019 to August 2021. Trained social workers obtained written informed consent with witness for the study from the participant. All the patients were provided supportive counselling for regular treatment adherence. Patients were also provided refreshments during interview.

# Results

A total of 556 TB patients were screened for the study of which 466 patients were found eligible and followed-up from baseline to endline (Table 1). A total of 4039 family and other social network contacts were listed by the eligible TB patients whom they considered as an important social member of his or her life.

## Attribute characteristics of TB patients and contacts

Background characteristic of the participant TB patients is provided in (Table 1). Majority of the participants were male (61.37%), aged less than 46 years (57.51%), school educated (62.88%), formally employed (45.06%) and had pulmonary TB (71.46%). Attribute characteristics of 4039 individuals who were listed as family and social network contacts of TB patients are provided in (Table 2). Majority of the contacts were aged less than 46 (64.17%), school educated (58.28%), formally employed (44.42%) and half of them were male (50.36%). In terms of contact type, majority were extended family and relative contacts (59%) followed by family members (24.71%), and friendship contacts (10.47%). Occupational and neighbourhood contacts were relatively less (≤3.05%).

## Disclosures made at different time points during treatment period

TB disclosure status of TB patients with their social network contacts at different time points from their treatment initiation period till their treatment completion is provided in (Table 3). Maximum disclosures were made with family members within 15 days of TB patient's

**Table 1. Background characteristics of TB patients.**

| Characteristics | | Frequency N = 466 | Percentage % |
|---|---|---|---|
| **Age in years** | Up to 30 | 130 | 27.90 |
| | 31–45 | 138 | 29.61 |
| | 46–55 | 108 | 23.18 |
| | >55 | 90 | 19.31 |
| **Gender** | Female | 180 | 38.63 |
| | Male | 286 | 61.37 |
| **Education** | Illiterate | 79 | 16.95 |
| | School Level | 293 | 62.88 |
| | Graduate Level | 94 | 20.17 |
| **Marital status** | Unmarried | 154 | 33.05 |
| | Married | 312 | 66.95 |
| **Occupation** | Daily wages | 96 | 20.60 |
| | Formal employed | 210 | 45.06 |
| | Unemployed | 160 | 34.33 |
| **Site of TB** | Extra Pulmonary TB | 133 | 28.54 |
| | Pulmonary TB | 333 | 71.46 |

**Table 2. Background characteristics of social network contacts of TB patients.**

| Characteristics | | Frequency N = 4039 | Percentage % |
|---|---|---|---|
| Age in years | Up to 30 | 1287 | 31.86 |
| | 31–45 | 1305 | 32.31 |
| | 46–55 | 762 | 18.87 |
| | >55 | 685 | 16.96 |
| Gender | Female | 2005 | 49.64 |
| | Male | 2034 | 50.36 |
| Education | Illiterate | 835 | 20.67 |
| | School Level | 2354 | 58.28 |
| | Graduate Level | 850 | 21.04 |
| Occupation | Daily wages | 479 | 11.86 |
| | Formally employed | 1794 | 44.42 |
| | Unemployed | 1766 | 43.72 |
| Contact type | Extended family/ Relatives | 2381 | 58.95 |
| | Family | 998 | 24.71 |
| | Friends | 423 | 10.47 |
| | Neighbor | 114 | 2.82 |
| | Occupational | 123 | 3.05 |

treatment initiation (92.88%). Almost half of the extended family/relatives, occupational contacts and friendship contacts (44–58%) were disclosed at treatment initiation. Disclosure with neighbourhood contacts were less with only one-fourth being disclosed at treatment initiation. Overall 64% of the social network contacts (extra-family members and rest) were disclosed about the TB status of patients at treatment initiation. Incremental disclosures made during the course of treatment at 60–90 days of treatment (corresponding with intensive treatment phase completion) was found to be highest among the friendship contacts (12.29%). Incremental disclosures made during the end phase of treatment (150–180 days) was found to be highest among the neighbourhood contacts (12.28%), followed by occupational contacts (11.38%) and friendship contacts (7%). Overall an additional 3.46% disclosures happened during the treatment period after the treatment initiation. The total cumulative disclosure from treatment initiation till treatment completion was predominantly made with family contacts (93.48%) followed by occupational contacts, extended family/relatives, and friends contacts (63.75%) and neighbourhood contacts (43.85%). The level of disclosures made by TB patients

**Table 3. Disclosure status of TB patients with their social network contacts at different time points.**

| Contact Type | Disclosure status <15 days of treatment initiation | | | Disclosure status 60–90 days of treatment initiation | | | Disclosure status 150–180 days of treatment initiation | | | Total disclosures from treatment initiation to treatment end | | | Z test |
|---|---|---|---|---|---|---|---|---|---|---|---|---|---|
| | N | % | 95% CI | N | % | 95% CI | N | % | 95% CI | N | % | 95% CI | |
| Family (N = 998) | 927 | 92.88 | 91.11–94.40 | - | - | | - | - | | 933 | 93.48 | 91.77–94.93 | 0.39 |
| Extended Family/ Relatives (N = 2381) | 1376 | 57.79 | 55.77–59.78 | 63 | 2.64 | 2.03–3.37 | 79 | 3.31 | 2.63–4.11 | 1518 | 63.75 | 61.78–65.68 | 0.00 |
| Friends (N = 423) | 185 | 43.73 | 38.94–48.61 | 52 | 12.29 | 9.31–15.80 | 30 | 7.09 | 4.83–9.96 | 267 | 63.12 | 58.32–67.73 | 0.00 |
| Neighbour (N = 114) | 27 | 23.68 | 16.22–32.55 | - | - | - | 14 | 12.28 | 6.87–19.74 | 50 | 43.85 | 34.58–53.46 | 0.00 |
| Occupational (N = 123) | 68 | 55 | 46.05–64.25 | - | - | - | 14 | 11.38 | 6.36–18.35 | 87 | 70.73 | 61.85–78.58 | 0.01 |
| Total (N = 4039) | 2583 | 63.95 | 62.44–65.43 | 132 | 3.26 | 2.74–3.86 | 140 | 3.46 | 2.92–4.07 | 2855 | 70.68 | 69.25–72.08 | 0.00 |

(-) denotes where N<10

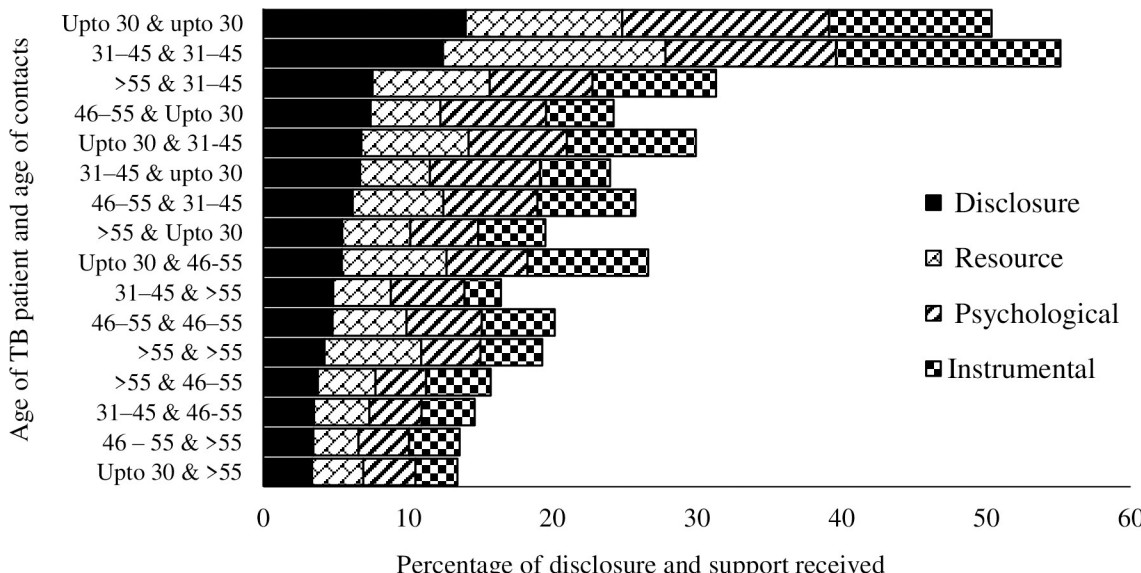

**Fig 1. Proportion of age specific TB disclosure happened between index and contacts and post disclosure positive outcomes.**

with their social network contacts was found to be significantly different between the time point of their treatment initiation and their treatment completion.

## Disclosures by age and gender of patients and contacts

Maximum disclosure have happened between the index and contacts who were young and middle aged (30 to 45 years) and minimum disclosures have happened between the young and middle aged index patients (30 to 55 years) and older contacts (>55 years). The proportion of support exchanged from contact to index across age group was found proportional to the disclosure level. There was a statistically significant association between age-group wise disclosure levels and age-group wise supports provided (Fig 1). It was found that male-patient to male-contact disclosure was higher (>30%) and female-patient to male-contact was lowest (18%). The proportion of support exchanged from contact to index across their gender group was found proportional to disclosure level. There was a significant association between gender wise disclosure level and gender wise resource and instrumental support provided (Fig 2).

## Background characteristics of patients and contacts associated with disclosure status

Bivariate analysis shows that disclosure status was found to be significantly different in terms of TB patients as well as their contact characteristics. Disclosures were found to be higher among young age and older patients (<30 and >55years), illiterates and among patients with pulmonary TB. Contacts who were young aged and daily wagers differed significantly with respect to the disclosure status made by their respective TB patients. Contacts also differed significantly in terms of their contact types (family and extra-family contacts) in being disclosed about TB by the patients (Table 4).

In terms of TB patient characteristics, univariate analysis showed that patients who were in the middle age (31–55 years) were less likely to disclose their TB status when compared to young age group (<30 years). Patients who were illiterate and patients who had pulmonary TB were found to be more likely to disclose their TB status when compared to more educated and

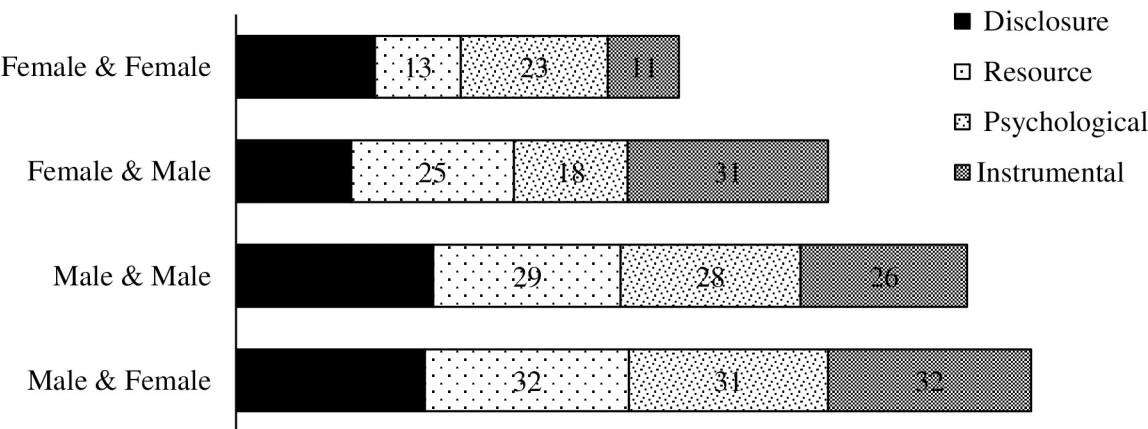

**Fig 2. Proportion of gender specific TB disclosure happened between index and contacts and post disclosure positive outcomes.**

extra-pulmonary TB patients respectively. In terms of contact characteristics of TB patients, univariate analysis showed that contacts who were in middle and old age (46 and above years) were less likely to get disclosed about their contact's TB status, when compared to younger age contacts (<30 years). Contacts who were less educated were less likely to get disclosed about TB status when compared to more educated contacts. Contacts who were engaged in informal works or daily labourer were more likely to get disclosed about TB when compared to unemployed contacts. In terms of contact type, all extra-familial contacts including extra household, friendship and neighbourhood contacts were less likely to get disclosed about TB when compared to family contacts (Table 5).

Multivariate regression analysis showed that TB patients who were in the age group of 31–45 years (AOR 0.56, 95% CI 0.44–0.69 p = 0.00), 46–55 years (AOR 0.46, 95% CI 0.36–0.59, p = 0.00) and above 55 years (AOR 0.74, 95% CI 0.57–0.97, p = 0.03 respectively) were less likely to disclose their TB status when compared to young age group (<30 years). Patients who were illiterate (AOR 3.91, 95% CI 2.82–5.43, p = 0.00) were almost four times more likely to disclose their TB status when compared to more educated TB patients. Similarly patients who had school level education were (AOR 1.58, 95% CI 1.26–1.98, p = 0.00) more likely to disclose their TB status when compared to more educated TB patients. In terms of contact characteristics, contacts who were middle aged (46–55 years) were less likely (AOR 0.8, 95% CI 0.64–1.01, p = 0.06) to get disclosed about TB, when compared to younger age contacts (<30 years). Contacts who were engaged in informal works or daily labourer were more likely (AOR 1.43, 95% CI 1.10–1.86, p = 0.01) to get disclosed about TB when compared to unemployed contacts. Contacts who were less educated were less likely (AOR 0.76, 95% CI 0.61–0.95, p = 0.02) to get disclosed about TB status when compared to more educated contacts. Extra household contacts (AOR 0.11, 95% CI 0.09–0.15, p = 0.00) friendship contacts (AOR 0.11, 95% CI 0.08–0.15, p = 0.00) and occupational contacts (AOR 0.02, 95% CI 0.10–0.26, p = 0.00) were less likely to get disclosed about TB when compared to family contacts (Table 5). The mean VIF for the variables included in the multiple regression model was calculated as 1.62.

## Post disclosure support received by patients from contacts

Among all disclosed contacts of TB patients, family contacts have mostly provided resource support for them (44.90%) followed by friends (18.72%) and extended family contacts

**Table 4. Background characteristics of TB patients by disclosure and not disclosure.**

| Characteristics | | Not disclosed | | Disclosed | |
|---|---|---|---|---|---|
| **Patients** | | N | % | N | % |
| Age in years | Up to 30 years | 289 | 25.49 | 845 * | 74.51 |
| | 31–45 years | 391 | 33.28 | 784 | 66.72 |
| | 46–55 years | 313 | 33.37 | 625 | 66.63 |
| | >55 years | 191 | 24.12 | 601 | 75.88 |
| Gender | Male | 701 | 28.93 | 1722 | 71.07 |
| | Female | 483 | 29.89 | 1133 | 70.11 |
| Education | Illiterate | 102 | 16.32 | 523 * | 83.68 |
| | School Level | 824 | 31.50 | 1792 | 68.50 |
| | Graduate Level | 258 | 32.33 | 540 | 67.67 |
| Marital status | Unmarried | 334 | 27.24 | 892 | 72.76 |
| | Married | 850 | 30.22 | 1963 | 69.78 |
| Occupation | Daily wages | 245 | 31.49 | 533 | 68.51 |
| | Formal employed | 540 | 29.28 | 1304 | 70.72 |
| | Unemployed | 399 | 28.16 | 1018 | 71.84 |
| Site of TB | Pulmonary TB | 785 | 28.21 | 1998 | 71.79 |
| | Extra Pulmonary TB | 399 | 31.77 | 857 | 68.23 |
| **Contact** | | | | | |
| Contact type | Extended Family | 863 | 36.25 | 1518 * | 63.75 |
| | Family | 65 | 6.51 | 933 | 93.49 |
| | Friends | 156 | 36.88 | 267 | 63.12 |
| | Neighbour | 64 | 56.14 | 50 | 43.86 |
| | Occupational | 36 | 29.27 | 87 | 70.73 |
| Age in years | Up to 30 | 328 | 25.49 | 959 * | 74.51 |
| | 31–45 | 363 | 27.82 | 942 | 72.18 |
| | 46–55 | 262 | 34.38 | 500 | 65.62 |
| | >55 | 231 | 33.72 | 454 | 66.28 |
| Gender | Male | 612 | 30.52 | 1393 | 69.48 |
| | Female | 572 | 28.12 | 1462 | 71.88 |
| Education | Illiterate | 234 | 28.02 | 601 * | 71.98 |
| | School Level | 726 | 30.84 | 1628 | 69.16 |
| | Graduate Level | 224 | 26.35 | 626 | 73.65 |
| Occupation | Daily wages | 106 | 22.13 | 373 * | 77.87 |
| | Formal employed | 573 | 31.94 | 1221 | 68.06 |
| | Unemployed | 505 | 28.60 | 1261 | 71.40 |

*p<0.05 for statistical test of association

(14.22%). Almost two-third of all disclosed contacts have provided emotional support for TB patients (>71%) and instrumental support was provided mostly by family members (35.36%) followed by occupational (11.49%) and extended family contacts (9.02%). There was no significant association between the type of contacts and the level of emotional support provided by the contacts to patients (Table 6).

## Discussion

Ours is the first comprehensive assessment of disclosure patterns of TB patients in India which could help in understanding the disclosure preferences of TB patients, how disclosure

**Table 5. Results of univariate and multivariate analysis of factor predictive of TB disclosure status.**

| Characteristics | | OR | P | 95% CI | AOR | P | 95% CI |
|---|---|---|---|---|---|---|---|
| **Patients** | | | | | | | |
| Age in years | Upton 30 | **Ref** | | | **Ref** | | |
| | 31–45 | 0.68 | 0.00 | 0.57–0.82 | 0.56 | 0.00 | 0.44–0.69 |
| | 46–55 | 0.68 | 0.00 | 0.56–0.82 | 0.46 | 0.00 | 0.36–0.59 |
| | >55 | 1.07 | 0.49 | 0.87–1.32 | 0.74 | 0.03 | 0.57–0.97 |
| Gender | Female | Ref | | | NA | | |
| | Male | 1.04 | 0.51 | 0.91–1.20 | | | |
| Education | Graduate | **Ref** | | | **Ref** | | |
| | Illiterate | 2.44 | 0.00 | 1.89–3.17 | 3.91 | 0.00 | 2.82–5.43 |
| | School | 1.03 | 0.66 | 0.87–1.23 | 1.58 | 0.00 | 1.26–1.98 |
| Occupation | Unemployed | **Ref** | | | NA | | |
| | Regular\Self Employed | 0.94 | 0.48 | 0.81–1.10 | | | |
| | Daily wages | 0.85 | 0.10 | 0.7–1.03 | | | |
| Marital Status | Unmarried | **Ref** | | | NA | | |
| | Married | 0.86 | 0.05 | 0.74–1.00 | | | |
| Type of TB | EPTB | **Ref** | | | **Ref** | | |
| Type of TB | PTB | 1.18 | 0.02 | 1.02–1.36 | 1.02 | 0.75 | 0.87–1.2 |
| **Contacts** | | | | | | | |
| Age in years | Upton 30 | **Ref** | | | **Ref** | | |
| | 31–45 | 0.88 | 0.18 | 0.74–1.05 | 1.15 | 0.16 | 0.95–1.41 |
| | 46–55 | 0.65 | 0.00 | 0.53–0.79 | 0.8 | 0.06 | 0.64–1.01 |
| | >55 | 0.67 | 0.00 | 0.54–0.82 | 0.87 | 0.27 | 0.67–1.11 |
| Gender | Female | **Ref** | | | NA | | |
| | Male | 0.89 | 0.09 | 0.70–1.01 | | | |
| Education | Graduate | **Ref** | | | **Ref** | | |
| | Illiterate | 0.91 | 0.44 | 0.74–1.13 | 0.83 | 0.22 | 0.63–1.11 |
| | School | 0.80 | 0.01 | 0.67–0.95 | 0.76 | 0.02 | 0.61–0.95 |
| Occupation | Unemployed | **Ref** | | | **Ref** | | |
| | Regular\ Self Employed | 0.85 | 0.03 | 0.73–0.98 | 0.91 | 0.26 | 0.77–1.07 |
| | Daily wages | 1.40 | 0.01 | 1.10–1.78 | 1.43 | 0.01 | 1.10–1.86 |
| Relationship | Family | **Ref** | | | **Ref** | | |
| | Extended Family | 0.12 | 0.00 | 0.09–0.15 | 0.11 | 0.00 | 0.09–0.15 |
| | Friends | 0.11 | 0.00 | 0.08–0.16 | 0.11 | 0.00 | 0.08–0.15 |
| | Neighbour | 0.05 | 0.00 | 0.03–0.08 | 0.02 | 0.00 | 0.03–0.07 |
| | Occupational | 0.16 | 0.00 | 0.11–0.27 | 0.16 | 0.00 | 0.1–0.26 |

NA refers to variables which were not included in multivariate regression due to statistical insignificant univariate analysis

unfolded over time and what was the usefulness of such disclosure for patients. This study had assessed the disclosure status of TB patients as a dynamic process which unfolds over the patient's treatment cascade and from a familial and broader social network perspective of the patients. With the advantage of having collected comprehensive information on the familial and other social network contacts of TB patients, the present study had estimated the relative disclosure proportion of TB patients based on the specific relationship types.

The findings show that TB patients disclosed their disease status predominantly to their family members within two weeks of their treatment initiation. Such immediate disclosure to family contacts remains important for two reasons. First is that such disclosures would benefit

**Table 6. Positive supports received by TB patients from their disclosed contacts.**

| Relationship Type of Index | Total Disclosed contacts | Contacts who provided resource support | | Contacts who provided Psychological Support | | Contacts who provided Instrumental Support | |
|---|---|---|---|---|---|---|---|
| | N | N | % | N | | N | % |
| Extended Family | 1518 | 216 | 14.22[#] | 1127 | 74.24 | 137 | 9.02 |
| Family | 933 | 419 | 44.90 | 685 | 73.41 | 330 | 35.36 |
| Friends | 267 | 50 | 18.72 | 192 | 71.91 | 37 | 13.85 |
| Neighbour | 50 | 05 | 10.00 | 37 | 74.00 | 01 | 2.00 |
| Occupational | 87 | 12 | 13.79 | 66 | 75.86 | 10 | 11.49 |
| P-value | | 0.00* | | 0.92 | | 0.00* | |

*Chi square test of association

[#] Proportions were calculated with total disclosure (N) as denominators.

the TB patient's family contacts for undertaking prevention measures when the patient is still infective [22]. Secondly patients might be able to receive the immediate and necessary support from the household members for coping-up with the treatment challenges. Studies have shown that the initial days of treatment were challenging for the patients when they face emotional disturbance, physical weakness and loss of wages than during the later phase when they get adopted to it [25]. The present study also identified that the TB patients mostly received the monetary, instrumental and psychological support from their family contacts after their disclosure.

It was also notable that seven percent of TB patients have not disclosed their disease status to any of their family members at any point of their treatment duration. While the non-disclosure proportion is relatively low, still the numbers in absolute terms could be significant for India where every year more than two million people are newly diagnosed of TB. This subset of non-disclosing individuals could be important in the context of household transmission of TB. Also this sub set of patients might lack the necessary supports from the family members and thus experience increased challenges during the treatment period [26, 27]. Overall, the immediate need for disclosure with the family members underscores the importance of family as a key social institution for the TB patients during his/her challenging treatment cascade [28].

The present study highlights that TB patients have disclosed their status to almost 50% of their relative, friends and occupational contacts within the first two weeks of treatment initiation. This finding is of importance, considering that patients tend to have opened up their disease status to considerable number of their extra-family contacts, overlooking the concerns of stigma or discrimination [29, 30].

But the disclosures made to neighborhood contacts of patients (with whom they shared or socialized) were found to be the lowest within the first two weeks, highlighting the stigma and fear of discrimination felt by patients especially in urban neighborhood settings in which our study was conducted. TB similar to other stigmatized infectious disease like HIV is less accepted in the neighborhoods and sometimes results in eviction and denial of household for the infected patients in India and similar countries [31]. The lowest non-disclosures with neighborhood contacts also highlight the higher risk faced by neighbor who are in close contact with the patients in terms of disease transmission, which has been highlighted in a recent study in the same city [32].

From this baseline information, the study had tracked the change in disclosure status through the treatment journey of patients. We found that disease disclosure had an incremental increase by 4–12% (among friends, neighbors and occupational contacts) at 60–90 days of treatment and by 7–12% during the last phase of treatment (150–180 days). Such trends

highlights the underlying tendency to disclose the disease status when the TB patient progressively adapts towards the treatment. Thus the confidence or willingness to disclose the disease to friends and social contacts could be attributed to the normalcy (in physical and mental status) achieved during the course of treatment. Especially incremental disclosures made to occupational and neighborhood contacts significantly increased by more than 10% at the end of treatment period (150–180 days). TB patients who mostly discontinued their jobs during the initial months of disease are known to re-engage in the subsequent months due to betterment of health and thus the spike in disclosure at this point could be explained. This phenomenon of normalcy could also explain the increased disclosure to neighborhood contacts.

Except for the disclosures made to family members, the cumulative disclosures made by the TB patient to other extra-family social network members was found to be higher than the disclosures made during the initial period. Overall disclosures were added incrementally towards the end of treatment and it was found to be statistically significant. These findings have implications for evolving patient centric services and support systems for TB patients [33, 34]. Patient support systems are nurtured by way of building patient support groups, community support groups and patient provider forums [35, 36]. In this background the key role of the family members of the TB patients is highlighted by the findings of this study, whose confidence and trust remains pre-dominant for the TB patients. Thus any patient support system for the TB patients could be developed in such way that it lies in complete synergy with family members and care givers. Patient centric strategies could incorporate family members of patients as a key stakeholder in devising interventions [37]. Similarly the involvement of friends and occupational contacts in the patient support system could be considered at the late stage of treatment. Considering the low level of disclosures made with the neighborhood contacts, emphasize on community mobilization and advocacy interventions to address stigma and discrimination would be important.

There are other important findings which emerge from the study with respect to the characteristics of TB patients which are predictive of TB disclosure. With increasing age of TB patients and contacts there was a decreased likelihood of disclosure. The reason could be that the older age patients might face more constraints and concerns about sharing their disease status owing to their strong familial, social and professional roles and identities than the younger patients. TB being a highly prevalent disease in India, holds major socio-economic consequences for the affected individual and family, which could have constrained the elder patients in terms of disclosure than the relatively younger individuals.

In terms of educational status, it was found that illiterate TB patients were found to have increased likelihood of disclosure and contacts who were moderately educated and were less likely to get disclosed by the TB patients. This could be explained by the fact that educational status of individuals relates to their prominent social interactions in their personal and social environment, unlike less educated individuals whose socialisation role could be limited and less consequential. Adding strength to this was the finding that those contacts who were engaged in informal works or daily labour were more likely to get disclosed about their contact TB patient's status. High social visibility and physical labour related demands might be conducive for the disease disclosure of affected TB patients when compared to those in organised or unemployed working conditions.

It was found that younger and middle aged TB patients tended to disclose more to their same age group and had resulted in gaining more supports underscoring the demographic and socially conducive environment available for them. Male patients disclosure resulted in more support from their male and female patient contacts when compared to female patients contacts. This underscores male patients relatively strong social position within family and social network relations.

Our study also found that among those who have disclosed TB status with their different social network contacts, it resulted in significant level of resources support. Almost half of the family member contacts of the patients provided the resources support followed by their friends' contacts. The provision of resources support for patient is especially important considering the association of TB with multidimensional poverty in India [38]. Emotional support was the predominantly received by the patients from their all type contacts [39, 40]. Instrumental support was less received support by TB patients from their social network contacts except for family member who provided it high. This highlights that the most important consequence of TB disease disclosure could be the emotional support needed for patients. TB being a highly consequential disease which affects the social and mental health status of TB patients. Such high level of social consequences places them at the receiving end of emotional support which is provided by all social network contacts including family members. Thus the involvement of family members and close network contacts as patient supporters during the treatment period could be a viable way for sustaining a patient centric care for patients. Overall the findings show that disclosures results in significant generation of support among the social network members. This underscores, that the close social networks of patients (friends and workplace) together with family members could play a key and predominant role in the patient treatment outcomes.

The findings of this study has implications and evidences for devising policy and interventions for addressing issues related to TB disclosure. Stigma reduction interventions for TB could factor in the disclosure patterns observed in this study. Family being the predominant relationship with whom maximum disclosure had happened and positive supports are gained. Stigma reduction intervention for TB could focus on family level stigma reduction to rationalize and consolidate family members involvement and build collective resilience for the patients. Alternatively, there is a need for identifying TB patients who had zero family disclosures and assist them with social support interventions. Another practical implication of this study is in the context of evolving patient centric interventions for TB patients and enabling them with necessary emotional, resources and practical support during their treatment cascade. The present findings have underscored the disclosure timings and preferred relationship of disclosure which could help to evolve the patient's support system alongside the disclosure process. The low level of disclosures made with the neighborhood contacts underscores the need for community mobilization and advocacy interventions at community level.

## Strengths and limitation of the study

This study had provided important learnings in terms of administering a disclosure assessment tool among TB patients which would be challenging otherwise. The study had provided learnings in terms of analyzing and interpreting the social network relationships and support systems of TB patients from a timeline perspective which is not available in India. The study findings need to be interpreted with limitation. Our study sample consisted of only adult drug-sensitive TB patient's and thus may not be generalizable for drug-resistant TB patients and patients who are adolescents and children. The study findings could be further extended to study disclosure among drug-resistant TB patients and could be based on different settings (like rural, tribal areas) to understand the contextual differences in TB disclosures.

## Conclusion

Our study had found that family level disclosures were predominant and the patients were also tending to disclose the disease over the treatment period to their other social network members as well. Further it was noted that emotional support was predominantly received by the

TB patients from all their family and social network members. Resources and instrumental support also emerged as the key outcomes of the disclosure. Low level of education was more predictive of disclosure and higher age was less predictive of disclosure. As highlighted by this study finding, disclosure of TB remains a continuous process and is predicted by patient and contact characteristics. The findings of this study could inform researchers, service providers and the patient support communities to best understand TB disclosure process and patterns and could help in developing interventions to facilitate TB disclosures in a beneficial way for TB patients.

## Acknowledgments

The authors also would like to thank the National Tuberculosis Elimination Program (NTEP) Staff of Greater Chennai Corporation, Tamil Nadu, India, for their extensive support in conducting this challenging study for the first time in India.

## Author Contributions

**Conceptualization:** Karikalan Nagarajan, Malaisamy Muniyandi, Lavanya Jeyabal.

**Data curation:** Karikalan Nagarajan, Senthil Sellappan, Srimathi Karunanidhi, Keerthana Senthilkumar, Bharathidasan Palani, Rajendran Krishnan.

**Formal analysis:** Karikalan Nagarajan, Malaisamy Muniyandi, Senthil Sellappan, Srimathi Karunanidhi, Rajendran Krishnan.

**Funding acquisition:** Karikalan Nagarajan.

**Investigation:** Senthil Sellappan, Srimathi Karunanidhi, Keerthana Senthilkumar, Bharathidasan Palani, Lavanya Jeyabal, Rajendran Krishnan.

**Methodology:** Karikalan Nagarajan, Malaisamy Muniyandi, Keerthana Senthilkumar, Bharathidasan Palani, Lavanya Jeyabal, Rajendran Krishnan.

**Project administration:** Karikalan Nagarajan.

**Supervision:** Malaisamy Muniyandi, Senthil Sellappan, Srimathi Karunanidhi, Keerthana Senthilkumar, Bharathidasan Palani, Lavanya Jeyabal, Rajendran Krishnan.

**Validation:** Srimathi Karunanidhi, Bharathidasan Palani, Lavanya Jeyabal.

**Writing – original draft:** Karikalan Nagarajan, Malaisamy Muniyandi.

**Writing – review & editing:** Karikalan Nagarajan, Malaisamy Muniyandi, Senthil Sellappan, Srimathi Karunanidhi, Keerthana Senthilkumar, Bharathidasan Palani, Lavanya Jeyabal, Rajendran Krishnan.

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
