## [Decision Letter · Decision Letter 0]

4 Oct 2022

PONE-D-22-14977A study on tuberculosis disease disclosure patterns and its associated factors and outcomes: A prospective observational study in Chennai, south IndiaPLOS ONE

Dear Dr. Malaisamy Muniyandi,

Thank you for submitting your manuscript to PLOS ONE. After careful consideration, we feel that it has merit but does not fully meet PLOS ONE’s publication criteria as it currently stands. Therefore, we invite you to submit a revised version of the manuscript that addresses the points raised during the review process.

We look forward to receiving your revised manuscript.

Kind regards,

Jayanta Kumar Bora, PhD

Academic Editor

PLOS ONE

2.Thank you for stating the following financial disclosure:

“This study was supported through funding from Ministry of Human Resource Development, Government of India by its Impactful Policy Research in Social Science (IMPRESS) Scheme through Indian Council of Social Sciences Research (ICSSR), New Delhi.”

3.Thank you for stating the following in the Funding Section of your manuscript:

“This study was supported through funding from Ministry of Human Resource Development, Government of India by its Impactful Policy Research in Social Science (IMPRESS) Scheme through Indian Council of Social Sciences Research (ICSSR), New Delhi.”

“This study was supported through funding from Ministry of Human Resource Development, Government of India by its Impactful Policy Research in Social Science (IMPRESS) Scheme through Indian Council of Social Sciences Research (ICSSR), New Delhi.”

4.Thank you for stating the following in your Competing Interests section: 

“None reported”

5.In your Data Availability statement, you have not specified where the minimal data set underlying the results described in your manuscript can be found. PLOS defines a study's minimal data set as the underlying data used to reach the conclusions drawn in the manuscript and any additional data required to replicate the reported study findings in their entirety. All PLOS journals require that the minimal data set be made fully available. For more information about our data policy, please see http://journals.plos.org/plosone/s/data-availability.

Reviewers' comments:

Reviewer's Responses to Questions

**Comments to the Author**

1. Is the manuscript technically sound, and do the data support the conclusions?

Reviewer #1: Yes

Reviewer #2: Yes

Reviewer #3: Partly

2. Has the statistical analysis been performed appropriately and rigorously? 

Reviewer #1: Yes

Reviewer #2: Yes

Reviewer #3: Yes

3. Have the authors made all data underlying the findings in their manuscript fully available?

Reviewer #1: Yes

Reviewer #2: Yes

Reviewer #3: No

4. Is the manuscript presented in an intelligible fashion and written in standard English?

Reviewer #1: Yes

Reviewer #2: Yes

Reviewer #3: Yes

5. Review Comments to the Author

Reviewer #1: Journal Name: PLOS

Manuscript ID: PONE-D-22-14977

Manuscript Title: “A study on tuberculosis disease disclosure patterns and its associated factors and outcomes: A prospective observational study in Chennai, south India"

Reviewer Comments:

The authors have investigated tuberculosis disease disclosure patterns and its associated factors. Reviewer has found many some problems. Detailed evaluation is given below:

Abstract: Authors should reduce abstract.

Background: Please state the research questions/hypothesis of the study.

Methodology:

1. Did authors measure internal consistency of their questionnaire, if yes how?

2. In Statistical analysis, authors should only describe statistical methods, i.e mention statistical techniques/models, and describe why they use in their study. Please remove outcome and independent variables from the subtitle and make a subtitle, Variable where authors describe about outcome and independent variables separately.

Authors used univariate and multivariate (multiple) logistic regression model, in multiple logistic regression model consider more than one independent variable, great chance to get multicollinearity problems among selected independent variables, please describe who authors check the multicollinearity problems, and how they overcome the problem if available.

3. Please describe separately inclusion/exclusion criteria, sampling, data collection procedure etc.

Reviewer #2: This is an interesting paper based on primary data.

The paper is well written . However it need improvement in following

1. In introduction the social stigma attached to TB in India may be discussed. Second, how the disclosure adversely/ positively affect patients treatment and health may be discussed.

2.From title Outcome may be dropped

3.Tables: All tables should have uniformly 2 decimals.

4. Table 3 No. should be replaced with N. Do not show number or % for N<30. there are many cases with N less than 10. Must be replaced with __ mentioning in footnote number not shown due to small number

Put N as last row (all cases)

% in parentthesese should be shown without barcket

It basically shown the percentage of XX by characteristics

5. Keep all column from Odd ratio onwards. Drop the columns except variable

6. A recent paper is missed in literaure review

Pathak, D., Vasishtha, G., & Mohanty, S. K. (2021). Association of multidimensional poverty and tuberculosis in India. BMC Public Health, 21(1), 1-12.

Reviewer #3: This paper provides very interesting research work addressing the prospective observational evidence of the TB diseases patterns in Chennai.

1. The introduction section requires substantial improvement. It looks too short and not enough existing studies included. I would suggest adding more literatures and add a conceptual framework.

2. Can you remove the word South India from the title (if possible).?

3. In Table 1, the total frequency has not been included? Can you revise table 1?

4. In Table 2, the total frequency has also not been included? Can you revise table 2?

5. The age-group needs to be reclassified, for example- 0-4 years, 5-14 years, 15-29 years, 30-44 years, 45-59 years, and 60+ years.

6. In Table 5, the adjusted odds ratio for occupation is missing? why? also why the reference category is not arranged properly?

7. The Table 6 is not clear.

8. The title of the Figure 1 is unclear.

9. In Table 4, you have use * sign. But the meaning of * sign is missing in the footnotes.

10. In the result section of the multivariate analysis, the interpretation needs substantial improvement. For-example. include the values of odds ratio with their p value or std.error or confidence interval.

11. What is the strength and drawback of this study?

12.Author should make the integers into one or two decimal places.

13. Any additional policy recommendation?

14. Write the data availability section separately in brief. For example... whether it is publicly available or not. if not then explain why?

6. PLOS authors have the option to publish the peer review history of their article (what does this mean?). If published, this will include your full peer review and any attached files.

Reviewer #1: No

Reviewer #2: **Yes: **Sanjay K Mohanty

Reviewer #3: **Yes: **Saddaf Naaz Akhtar

---

## [Author Response · Author response to Decision Letter 0]

7 Nov 2022

S No Reviewer’s Comments Authors responses Page No in Manuscript

 Reviewer 1 

1 Authors should reduce abstract. As suggested we have edited and reduced abstract to nearly 300 words. Page -2

2 Please state the research questions/hypothesis of the study. As suggested we have added research question under the method section of the revised manuscript. Page 5

3 Did authors measure internal consistency of their questionnaire, if yes how? We have used a test -retest method and certain other checks to ensure the internal consistency of the tool which we have used. We have explained this in the method section and have added a published reference which we have made already. Page 8

4 In Statistical analysis, authors should only describe statistical methods, i.e mention statistical techniques/models, and describe why they use in their study. 

Please remove outcome and independent variables from the subtitle and make a subtitle, Variable where authors describe about outcome and independent variables separately. As suggested by reviewer we have revised and separated the statistical methods and outcome/ independent variables under different subtitles. Page 8

5 Authors used univariate and multivariate (multiple) logistic regression model, in multiple logistic regression model consider more than one independent variable, great chance to get multicollinearity problems among selected independent variables, please describe who authors check the multicollinearity problems, and how they overcome the problem if available. We thank the reviewer for raising this important point. For multiple regressions we calculated variance inflation factor VIA ins STATA and found that all variables had less than 4 and the mean VIA was only 1.63 indicating that that collinearity was not affecting our analysis. We have included this detail in the revised manuscript I results section.

 Page 8,12

6 Please describe separately inclusion/exclusion criteria, sampling, data collection procedure etc. As suggested have given separate sub titles and details of inclusion/exclusion criteria, sampling, data collection procedure, analysis, and outcome/ independent variables in the revised manuscript. Page 5

 Reviewer 2 Comments 

1 This is an interesting paper based on primary data. The paper is well written . However it need improvement in following We thank the reviewer for positively acknowledging the importance of our study. We have made the necessary changes as suggested 

2 In introduction the social stigma attached to TB in India may be discussed. Second, how the disclosure adversely/ positively affect patients treatment and health may be discussed. As suggested by the reviewers we have outlined the relationship between disclosure, stigma, poor mental status etc with references in the revised papers. We have added the available literature on positive outcomes of TB disclosure from a setting similar to India. We have also provided a detailed discussion of how disclosures and positive outcomes are important Page 3,4

3 .From title Outcome may be dropped We have modified the title as suggested by the reviewer Page 1

4 .Tables: All tables should have uniformly 2 decimals. As suggested we have given 2 decimals uniformly in all tables.

 Page 22-27

5 Table 3 No. should be replaced with N. Do not show number or % for N<30. there are many cases with N less than 10. Must be replaced with __ mentioning in footnote number not shown due to small number

Put N as last row (all cases) % in parentheses should be shown without bracket It basically shown the percentage of XX by characteristics

 As suggested we have replaced the lower numbers . however we have kept it at a level less than <10. This was to highlight incremental changes in disclosures made at different timepoints which are important 

We have removed the parenthesis for % in all tables and have made N %as the last row in table header Page 24

6 Keep all column from Odd ratio onwards. Drop the columns except variable We have dropped the columns before Odds ratio as suggested. 

 Page 22-27

7 A recent paper is missed in literaure review

Pathak, D., Vasishtha, G., & Mohanty, S. K. (2021). Association of multidimensional poverty and tuberculosis in India. BMC Public Health, 21(1), 1-12. As suggested we have added this recent and important literature in the revised manuscript Page 15

 Reviewer 3 Comments 

1 This paper provides very interesting research work addressing the prospective observational evidence of the TB diseases patterns in Chennai. We thank the reviewer for positively acknowledging the importance of our study. 

2 The introduction section requires substantial improvement. It looks too short and not enough existing studies included. I would suggest adding more literatures and add a conceptual framework. As suggested by the reviewers we have improved our introduction section and have added the literature. We have outlined the relationship between disclosure, stigma, poor mental status etc with references in the revised papers. Page 3-4

3 Can you remove the word South India from the title (if possible).? We have modified the title as suggested by the reviewer Page 1

4 In Table 1, the total frequency has not been included? Can you revise table 1? We have added the total frequency in Table 1 as suggested Page 22

5 In Table 2, the total frequency has also not been included? Can you revise table 2? We have added the total frequency in Table 2 as suggested Page 23

6 The age-group needs to be reclassified, for example- 0-4 years, 5-14 years, 15-29 years, 30-44 years, 45-59 years, and 60+ years. We would like to clarify that the present study included only adult TB patients and their contacts. As such our age categorisation were made from 18 years and above. Our categorisation thus does not have 0-4, 5-14 and 15 plus years. Our categorisation was made based on the prior assumption of TB burden among different age brackets based on TB program data in the study settings. 

7 In Table 5, the adjusted odds ratio for occupation is missing? why? also why the reference category is not arranged properly? Variables which were statistically insignificant IN univariate analysis was not included in multivariate regression. We have added a footnotes to clarify this. Also we have edited the tables and made clearer the reference category and rest. Page 26

8 The Table 6 is not clear. We have edited the Table 6 title, headers and given footnotes to make it clearer. Page 27

9 The title of the Figure 1 is unclear. We have edited the title of figure to make it clearer. Page 28

10 In Table 4, you have use * sign. But the meaning of * sign is missing in the footnotes. We have added the meaning of * sign in the revised manuscript. Page 25

11 In the result section of the multivariate analysis, the interpretation needs substantial improvement. For-example. include the values of odds ratio with their p value or std.error or confidence interval. As suggested we have added the values of odds ratio, p value and confidence interval and added explanations in the results part. Page 11

12 What is the strength and drawback of this study? We have added strengths and drawbacks of this study in the discussion part as suggested by the reviewer Page 16,17

13 Author should make the integers into one or two decimal places. As suggested we have standardised the decimals to two integers Page 22-27

14 Any additional policy recommendation? We have added policy recommendations in the discussion part as suggested by the reviewer Page 16

15 Write the data availability section separately in brief. For example... whether it is publicly available or not. if not then explain why? We have added the data availability section as suggested by the reviewer Page 18

---

## [Decision Letter · Decision Letter 1]

15 Dec 2022

PONE-D-22-14977R1A study on tuberculosis disease disclosure patterns and its associated factors: Findings from a prospective observational study in ChennaiPLOS ONE

Dear Dr. Malaisamy Muniyandi,

Thank you for submitting your manuscript to PLOS ONE. After careful consideration, we feel that it has merit but does not fully meet PLOS ONE’s publication criteria as it currently stands. Therefore, we invite you to submit a revised version of the manuscript that addresses the points raised during the review process.

We look forward to receiving your revised manuscript.

Kind regards,

Jayanta Kumar Bora, PhD

Academic Editor

PLOS ONE

Journal Requirements:

Reviewers' comments:

Reviewer's Responses to Questions

**Comments to the Author**

1. If the authors have adequately addressed your comments raised in a previous round of review and you feel that this manuscript is now acceptable for publication, you may indicate that here to bypass the “Comments to the Author” section, enter your conflict of interest statement in the “Confidential to Editor” section, and submit your "Accept" recommendation.

Reviewer #1: All comments have been addressed

Reviewer #3: All comments have been addressed

2. Is the manuscript technically sound, and do the data support the conclusions?

Reviewer #1: Yes

Reviewer #3: Yes

3. Has the statistical analysis been performed appropriately and rigorously? 

Reviewer #1: Yes

Reviewer #3: Yes

4. Have the authors made all data underlying the findings in their manuscript fully available?

Reviewer #1: Yes

Reviewer #3: Yes

5. Is the manuscript presented in an intelligible fashion and written in standard English?

Reviewer #1: Yes

Reviewer #3: Yes

6. Review Comments to the Author

Reviewer #1: Authors have addressed all comments. However, I have three comments/suggestions;

Methodology

1. Please move “Research question” from Methodology to Background (Please follow the paper, https://reader.elsevier.com/reader/sd/pii/S2666915321000962?token=84C91341547EA36C337EA4417603D907AE4D4853F0B66663B86A1BDC476A79390C4E01EA421AF721AE46FF91A511EF8D&originRegion=eu-west-1&originCreation=20221208125421)

2. In “Exclusion Criteria” Please complete the sentence “Drug resistant TB patients, TB patients with HIV and other life-threatening comorbidity”

3. Please provide the mathematical formula that was used for calculating sample size.

Reviewer #3: I am thankful to the authors for revising and updating the manuscript with my comments. I don't have additional comments.

7. PLOS authors have the option to publish the peer review history of their article (what does this mean?). If published, this will include your full peer review and any attached files.

Reviewer #1: No

Reviewer #3: **Yes: **Saddaf Naaz Akhtar

---

## [Author Response · Author response to Decision Letter 1]

5 Jan 2023

Reviewer 1

1. Authors have addressed all comments. However, I have three comments/suggestions;

 We thank the reviewer for acknowledging the revision.

2. Please move “Research question” from Methodology to Background 

 As suggested we have added research question under the background section of the revised manuscript as a separate para

3. In “Exclusion Criteria” Please complete the sentence “Drug resistant TB patients, TB patients with HIV and other life-threatening comorbidity”

 We have reworded this sentence 

4. Please provide the mathematical formula that was used for calculating sample size

 As suggested we have provided the formula used for sample size in the revised version

Reviewer 3 Comments

1. I am thankful to the authors for revising and updating the manuscript with my comments. I don't have additional comments.

 We thank the reviewer for acknowledging the revision

---

## [Editor Report · Decision Letter 2]

10 Jan 2023

A study on tuberculosis disease disclosure patterns and its associated factors: Findings from a prospective observational study in Chennai

PONE-D-22-14977R2

Dear Dr. Malaisamy Muniyandi,

We’re pleased to inform you that your manuscript has been judged scientifically suitable for publication subject to address the minor comments raised by one reviewer. Once it is done then it will be formally accepted for publication and should meets all outstanding technical requirements.

Kind regards,

Jayanta Kumar Bora,PhD

Academic Editor

PLOS ONE
---

## [Editor Report · Acceptance letter]

17 Jan 2023

PONE-D-22-14977R2 

A study on tuberculosis disease disclosure patterns and its associated factors: Findings from a prospective observational study in Chennai 

Dear Dr. Muniyandi:

I'm pleased to inform you that your manuscript has been deemed suitable for publication in PLOS ONE. Congratulations! Your manuscript is now with our production department. 

Kind regards, 

on behalf of

Dr. Jayanta Kumar Bora 

Academic Editor

PLOS ONE